# NexusFlow:
# Unifying Disparate Tasks under Partial Supervision via Invertible Flow Networks

## Abstract

Partially Supervised Multi-Task Learning (PS-MTL) aims to leverage knowledge across tasks when annotations are incomplete. Existing approaches, however, have largely focused on the simpler setting of homogeneous, dense prediction tasks, leaving the more realistic challenge of learning from structurally diverse tasks unexplored. This paper addresses this critical gap by introducing **NexusFlow**, a novel, lightweight, and plug-and-play framework. We establish a challenging new benchmark where supervision for the highly disparate tasks of dense map reconstruction and sparse multi-object tracking is split across different geographic domains, compounding task heterogeneity with a significant domain gap. Nexus-Flow introduces a pair of surrogate networks with invertible coupling layers to align the latent feature distributions of these tasks, creating a unified representation that enables effective knowledge transfer. We validate our framework's effectiveness on these core perception tasks in autonomous driving, demonstrating state-of-the-art results on the nuScenes benchmark. Our approach significantly outperforms strong partially supervised baselines. **Our code and video demos are available in the supplementary material.**

## 1 Introduction

Learning multiple tasks simultaneously via Multi-Task Learning (MTL) is a powerful paradigm for improving model efficiency and generalization while avoiding redundant training Xu et al. (2018); Zhang et al. (2019); Vandenhende et al. (2020); Li et al. (2024). However, its real-world applicability is often limited by the prohibitive cost of acquiring exhaustive annotations for every task, particularly in vision-heavy domains. In practice, datasets frequently lack labels for some tasks, or contain incomplete and unreliable annotations, motivating the crucial research direction of Partially Supervised Multi-Task Learning (PS-MTL) Li et al. (2022). Significant progress has been made when tasks are homogeneous dense predictions, thanks to their natural interdependence Zamir et al. (2018; 2020). This has enabled joint training in both partially and fully supervised settings for tasks such as semantic segmentation, depth estimation, and surface normal prediction Eigen et al. (2014); He et al. (2017); Chen et al. (2018); Poggi et al. (2020); Zhang et al. (2020). Yet, when tasks are structurally disparate (e.g., one requiring dense pixel-wise labels and another producing sparse instance-level outputs), the challenge becomes far greater and has received very limited attention. This difficulty is further compounded in realistic scenarios like autonomous driving, where supervision for each task comes from different geographic domains, introducing both structural disparity and domain shift into the training setup.

**Problem statement.** The conventional paradigm of MTL aims to improve generalization by learning shared representations across multiple tasks Zhang et al. (2019). This paradigm often assumes an idealized scenario where full ground-truth annotations for all tasks are simultaneously available for joint training. In practice, however, large-scale datasets rarely contain comprehensive labels for every task of interest, giving rise to the setting of Partially Supervised Multi-Task Learning (PS-MTL) Li et al. (2022). Most existing work on PS-MTL has focused on the relatively simple case of homogeneous dense prediction tasks, such as semantic segmentation and depth estimation Li et al. (2024). In this setting, partially annotated data is usually generated by randomly dropping task labels on a per-sample basis, which allows effective knowledge transfer under controlled conditions.

Yet, this scenario underestimates the real-world difficulty of PS-MTL Ye & Xu (2024). In practice, a much harder case frequently arises: each task is labeled only in its specific (geographic) region or domain. This introduces not only structural disparity between tasks (e.g., dense mapping versus sparse multi-object tracking) but also a substantial domain gap between supervision sources. Despite being a common challenge in real-world applications, this setting remains largely unexplored. To the best of our knowledge, we are the first to systematically address PS-MTL across fundamentally different tasks under such domain-partitioned supervision. Moreover, we take on this challenge in the complex and safety-critical domain of autonomous driving, tackling the highly diverse objectives of static map reconstruction and dynamic multi-object tracking.

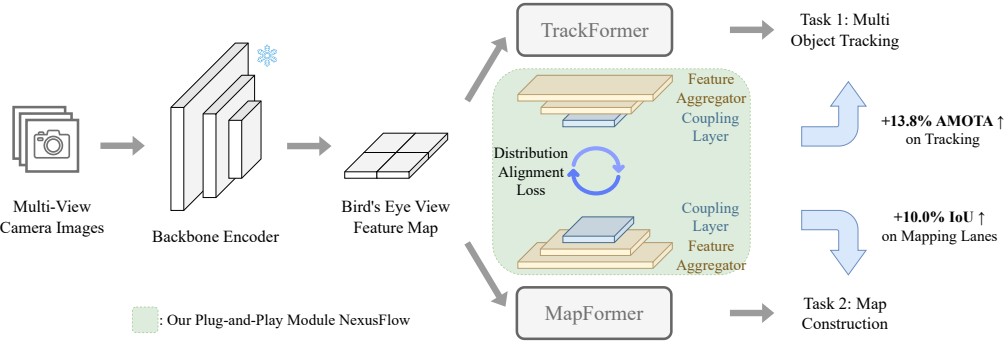

Figure 1: Our simple, plug-and-play module dramatically boosts the tracking and mapping performance from the UniAD Hu et al. (2023) baseline.

**Motivation.** The motivation for our work stems from the unique and formidable challenges of multi-task perception in autonomous driving Wang et al.; Xing et al. (2025); Wang et al. (2025). Our framework is designed to simultaneously address two highly complex and fundamentally disparate objectives. ❶ Multi-object tracking (MOT), which requires end-to-end detection and temporal association of dynamic instances without relying on non-differentiable post-processing. ❷ Map reconstruction, where sparse queries are used to generate a structured representation of the road, capturing both "things", such as lanes and dividers, and "stuff", such as the drivable area. These two tasks differ not only in their output structure (sparse instance sets vs. dense geometric maps) but also in their learning dynamics. The challenge is further amplified in a partially supervised regime. Unlike prior PS-MTL settings where missing labels are randomly distributed across samples, in autonomous driving it is common for one task to be annotated only in specific regions (e.g., mapping in Boston, tracking in Singapore). This setting couples structural disparity with domain shift, making joint optimization especially difficult. Simply adapting methods designed for homogeneous dense tasks, such as semantic segmentation and depth estimation, proves infeasible and inefficient in this context. This leads us to the central question of our work:

*How can we enable effective knowledge transfer between fundamentally different tasks with incomplete supervision, in a lightweight and plug-and-play manner?*

**Approach.** In light of the immense complexity of jointly learning structurally disparate tasks, our proposed NexusFlow focuses on aligning latent feature distributions during standard backpropagation in a simple yet effective manner. We formulate the Partially Supervised Multi-Task Learning (PS-MTL) problem using a compact alignment module inspired by principles of flow-based invertible models Ardizzone et al. (2018); Behrmann et al. (2019); Kobyzev et al. (2020). Specifically, NexusFlow introduces independent invertible coupling layers that project the latent features from different tasks into a canonical space where their distributional differences can be explicitly minimized. The choice of invertible coupling layers is deliberate. Unlike conventional CNN-based feature alignment, coupling layers provide a bijective mapping that preserves all task-relevant information while enabling flexible transformation. This one-to-one mapping nature allows NexusFlow to align feature distributions without collapsing them, maintaining information capacity even when tasks differ drastically in structure (e.g., dense maps vs. sparse object tracks). At the same time, the lightweight design avoids the topological constraints and computational overhead of training heavy deep neural networks. The essence of NexusFlow is to unify the feature spaces of disparate tasks so that knowledge from one

can benefit the other, even when supervision is incomplete and domain-partitioned. This alignment not only reduces distributional discrepancy but also strengthens downstream task performance and fosters more robust representations. We demonstrate its effectiveness through multi autonomous driving perception tasks, theoretical and piratical analysis.

**Contributions.** In summary, our key contributions are as follows:

- To the best of our knowledge, we are the *first* to systematically formulate and address the challenging problem of Partially Supervised Multi-Task Learning (PS-MTL) for structurally disparate tasks under domain-partitioned supervision. We identify this as a critical yet largely unexplored frontier, particularly for complex perception systems like autonomous driving.
- We propose **NexusFlow**, a novel, lightweight, and plug-and-play framework that tackles this challenge. Our key innovation is the use of **independent invertible coupling layers** to align task-specific feature distributions, enabling effective cross-task knowledge transfer.
- We provide a theoretical and practical analysis, demonstrating that our use of invertible transformations provides a **provable guarantee** for aligning the feature distributions of disparate tasks.
- We conduct extensive experiments to validate the effectiveness and generality of NexusFlow. Our approach achieves new state-of-the-art (SOTA) results on the challenging nuScenes Caesar et al. (2020) benchmark for both multi-object tracking and map reconstruction under PS-MTL.

## 2 RELATED WORK

**Partially Supervised Multi-Task Learning.** Partially Supervised Multi-Task Learning (PS-MTL) tackles the challenge of learning multiple tasks when only a subset has labels per sample. Early approaches showed success mainly in **homogeneous dense prediction tasks** (e.g., semantic segmentation, depth estimation), using strategies like consistency regularization, pseudo-labeling, or adversarial training. For example, adversarial discriminators have been used to align distributions across partially labeled datasets Wang et al. (2022b), while consistency-based methods either employ model augmentation to provide pseudo-supervision Spinola et al. (2023) or regularize cross-task relationships in a shared space Li et al. (2022). Pseudo-labeling has also been advanced via hierarchical task tokens for "label discovery," enabling dense supervision for unlabeled tasks Zhang et al. (2024).

Despite these advances, such techniques remain tailored to homogeneous tasks and are not directly applicable to **structurally heterogeneous settings** like object tracking versus map segmentation. Even recent unified frameworks for autonomous driving Hu et al. (2023); Huang et al. (2023) assume full supervision, leaving unresolved the realistic case of partially annotated, structurally disparate tasks—where annotation costs are prohibitive Dulac-Arnold et al. (2021); Vettoruzzo et al. (2024); Dai et al. (2025). A central challenge is learning shared representations without negative transfer. Architectural methods explore optimal layer sharing Ruder et al. (2019), while alignment-based methods project features into a common latent space. Early pairwise approaches Li et al. (2022) struggled with quadratic complexity, motivating more scalable solutions like JTR Nishi et al. (2024), which stacks all predictions into a unified joint-task space, or StableMTL Cao et al. (2025), which applies a unified latent loss with efficient 1-to-N attention.

Beyond scalability, newer methods seek richer cross-task knowledge transfer. Region-aware strategies use SAM Kirillov et al. (2023) to detect local regions and model their features as Gaussian distributions, enabling fine-grained alignment Li et al. (2024). DiffusionMTL Ye & Xu (2024) reframes partially labeled outputs as noisy predictions and refines them through a denoising diffusion process with multi-task conditioning. These works highlight the trend towards more sophisticated, scalable PS-MTL approaches, but none address the harder setting of **structurally disparate tasks under domain-partitioned supervision**, which is the focus of our work.

**Multi-Task Learning for Autonomous Driving Perception.** Autonomous driving perception has shifted from modular pipelines (separate detection, tracking, and mapping) to integrated multi-task learning (MTL) frameworks, motivated by efficiency, performance, and shared representations.

A major breakthrough is end-to-end models that unify perception, prediction, and planning. UniAD Hu et al. (2023) pioneered this direction with a Bird's-Eye-View (BEV) representation and Transformer-based modules for tracking (TrackFormer), mapping (MapFormer), motion prediction (MotionFormer), and occupancy prediction (OccFormer), all linked by a query-based mechanism.

Building on this, GenAD Zheng et al. (2024) models the scene generatively for joint prediction and planning, while DriveTransformer Jia et al. (2025) parallelizes tasks via shared attention for scalability. VAD Jiang et al. (2023) instead uses a fully vectorized scene representation, improving efficiency and reducing collision rates.

Another key challenge is **domain generalization**, as perception models face shifts in weather, lighting, sensors, and geography. Wang et al. (2022a) highlight this as central to real-world ML. For BEV-based systems, Wang et al. (2023) analyze domain gaps in 3D detection and propose robust depth learning. Jiang et al. (2024) introduce DA-BEV for unsupervised adaptation, combining image-view and BEV features. Chang et al. (2024) unify domain generalization and adaptation with multi-view overlap depth constraints. Finally, the **cost of annotation** remains prohibitive: datasets are often partially labeled. Li Li et al. (2022) address this by mapping task pairs into a joint space to enable sharing under incomplete supervision, a critical issue for large-scale autonomous driving systems where exhaustive labels are impractical.

## 3 APPROACH

### 3.1 PRELIMINARIES

**Notations & Problem Definition.** We formulate our problem within the framework of Partially Supervised Multi-Task Learning (PS-MTL). We consider a set of $K$ tasks, denoted by $\mathcal{T} = \{\mathcal{T}_1, \ldots, \mathcal{T}_K\}$. In this work, we focus on the practical and challenging case of two tasks ($K = 2$). Our training dataset is given by $\mathcal{S} = \{(x_i, \mathbf{y}_i, \mathbf{m}_i)\}_{i=1}^N$, consisting of $N$ samples. For each sample $i$, $x_i$ is the input data (multi-view frames) and $\mathbf{y}_i = \{y_i^1, y_i^2\}$ is the complete set of potential ground-truth labels for the two tasks. The availability of labels is governed by a binary mask vector $\mathbf{m}_i = (m_i^1, m_i^2)$. Notably, we operate under a **strict partial supervision setting** where each sample is labeled for exactly one task in one geographic domain (Either Singapore or Boston). A key distinction in our problem formulation lies in how the supervision masks $\mathbf{m}_i$ are assigned. In conventional PS-MTL for dense prediction tasks, masks are typically generated randomly per sample Ruder et al. (2019); Nishi et al. (2024); Cao et al. (2025). This randomization naturally helps to bridge the information gap, as similar inputs across the dataset are likely to receive labels for different tasks, yielding a richer joint training signal. In contrast, our setting introduces a far more challenging scenario by assigning supervision masks according to **geographically distinct subsets of data**. Specifically, in our experiments on the nuScenes dataset Caesar et al. (2020), one task is annotated only in Boston scenes, while the other is annotated only in Singapore scenes. This creates a substantial **domain gap** between supervision sources, which, when combined with the inherent structural disparity of the tasks, dramatically increases the difficulty of effective knowledge transfer. The core challenge we address lies in the **structural disparity** between our two tasks: Multi-Object Tracking ($\mathcal{T}_{track}$) and Map Reconstruction ($\mathcal{T}_{map}$). This disparity is most evident in their output spaces:

- **Map Reconstruction** ($\mathcal{T}_{map}$): a **dense, grid-based representation**, $y^{map} \in \mathbb{R}^{H \times W \times C}$, where each spatial location is assigned a semantic class (e.g., lane, divider, drivable area).
- **Multi-Object Tracking** ($\mathcal{T}_{track}$): a **sparse, instance-level set**, $y^{track} = \{(\mathbf{b}_j, id_j)\}_{j=1}^M$, where each object is represented by a bounding box $\mathbf{b}_j$ and an identity $id_j$, with object number $M$ varing across samples.

Our objective is to train a **single unified model** $f_\theta$ that jointly predicts both the dense map and the sparse object set. The model must learn a **shared representation** that bridges the structural gap between these disparate tasks, while operating under partial supervision.

### 3.2 NEXUSFLOW: UNIFIED LATENT DISTRIBUTION FOR PARTIALLY SUPERVISED LEARNING

Our approach follows a two-phase training strategy: a pre-training phase followed by a fine-tuning phase, where our proposed module, NexusFlow can be introduced to enable cross-task knowledge transfer in both phases.

**Training Paradigm.** We start with a baseline perception model Hu et al. (2023) composed of a shared BEV (Bird's-Eye-View) feature extractor and two task-formers for mapping and tracking.

*Phase 1: Pre-training.* The model is trained end-to-end under the defined PS-MTL protocol, using only the available labels for each sample. This phase yields a well-initialized, pre-trained model in which the task-formers have learned strong, albeit not explicitly aligned, shared representations.

*Phase 2: Fine-tuning with NexusFlow.* We freeze the BEV encoder and fine-tune the task-formers. We then insert our lightweight NexusFlow module to align the latent distributions of the two tasks. The model is optimized jointly with standard task losses (from partial labels) and our proposed alignment loss.

As illustrated in Figure 2, Phase 1 pre-training captures only a limited amount of task-relevant information for both tasks $T_1$ and $T_2$. In Phase 2, fine-tuning with NexusFlow seeks to expand the shared task-relevant subspace while simultaneously suppressing task-irrelevant features (white).

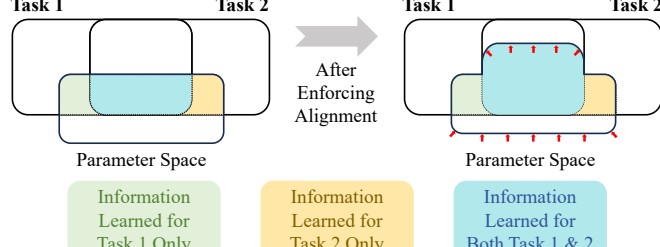

Figure 2: Illustration of the learning process of the NexusFlow. We aim to capture more shared information between the two tasks.

**NexusFlow Module Architecture.** As shown in Figure 1, NexusFlow is plugged in across the two task-formers Carion et al. (2020); Zeng et al. (2022); Zhang et al. (2022). NexusFlow consists of two identical surrogate networks, $S_{\text{surro}_1}(\cdot)$ and $S_{\text{surro}_2}(\cdot)$, that process the intermediate feature in parallel without altering the main path to the task-formers. Each surrogate has two components:

*Feature Aggregator:* From each task-former, its intermediate features are first compressed by a feature aggregator, $g'_{track}(\cdot)$ for tracking and $g'_{map}(\cdot)$ for mapping, into a fixed-dimension embedding. It is implemented by deformable transformer Zhu et al. (2020) $g'(\cdot)$, which efficiently aggregates salient spatial information into feature vector $f'_{map}$ and $f'_{track}$.

*Invertible Transformation:* The embedding $f'$ is then passed through an invertible coupling layer $\mathcal{C}(\cdot)$, a core component of normalizing flows Ardizzone et al. (2018); Behrmann et al. (2019); Kobyzev et al. (2020). In this work, we employ only the forward affine transformation $\mathcal{C}$ and use the second partition of the resulting embedding. Specifically, the coupling layer acts as:

$$\mathcal{C}(\mathbf{f}') = \big( \mathbf{f}'_2 \odot \exp\big(s(\mathbf{f}'_1)\big) + t(\mathbf{f}'_1)\big), \tag{1}$$

where the input $\mathbf{f}'$ is split into $(\mathbf{f}'_1, \mathbf{f}'_2)$, and both the scale $s(\cdot)$ and translation $t(\cdot)$ functions are implemented by multilayer perceptrons (MLPs). This layer performs a bijective transformation, mapping the feature into a canonical latent space, $z = \mathcal{C}(f')$, while preserving its information content.

**Distribution Alignment Objective.** A key advantage of our plug-and-play design is that $\mathcal{L}_{align}$ is added as an auxiliary objective. The baseline's original task-specific losses, $\mathcal{L}_{\text{map}}$ and $\mathcal{L}_{\text{track}}$, remain entirely unchanged:

$$\mathcal{L}_{\text{track}} = \lambda_{\text{focal}}\mathcal{L}_{\text{focal}} + \lambda_{l_1}\mathcal{L}_{l_1}, \tag{2}$$

$$\mathcal{L}_{\text{map}} = \lambda_{\text{focal}}\mathcal{L}_{\text{focal}} + \lambda_{l_1}\mathcal{L}_{l_1} + \lambda_{\text{iou}}\mathcal{L}_{\text{iou}} + \lambda_{\text{dice}}\mathcal{L}_{\text{dice}}. \tag{3}$$

The primary goal of NexusFlow is to enforce consistency between the transformed representations of the two tasks. The $f'_{map}$ and $f'_{track}$ are passed through the respective surrogate networks to obtain the latent variables $z_{map}$ and $z_{track}$:

$$z_{\text{map}} = \mathcal{C}_{\text{map}}\big(g'_{\text{map}}(f'_{map})\big), \quad z_{\text{track}} = \mathcal{C}_{\text{track}}\big(g'_{\text{track}}(f'_{track})\big). \tag{4}$$

We then introduce a simple but effective distribution alignment loss, $\mathcal{L}_{align}$, which minimizes the L2 distance between these two latent features. This effectively pulls their distributions closer together in the latent feature space, providing a better alignment:

$$\mathcal{L}_{\text{align}} = \|z_{\text{map}} - z_{\text{track}}\|_2^2. \tag{5}$$

The total loss of the training is:

$$\mathcal{L}_{\text{all}} = \mathcal{L}_{\text{map}} + \mathcal{L}_{\text{track}} + \lambda\mathcal{L}_{\text{align}}, \tag{6}$$

where $\lambda$ is a tuning-free hyperparameter: $\lambda = 1$.

Here, although our main focus is the two-phase fine-tuning strategy, NexusFlow can also be applied in a simpler one-phase joint training scheme, where $\mathcal{L}_{\text{align}}$ is active from the beginning. As shown in our experiments, this joint training approach also yields significant improvements, with only minor performance differences compared to the fine-tuning strategy, demonstrating the robustness and plug-and-play nature of NexusFlow.

### 3.3 THEORETICAL ANALYSIS

We now provide a formal analysis of our alignment mechanism, whose effectiveness hinges on a key property of the coupling layers $\mathcal{C}(\cdot)$: their invertibility. This property guarantees a one-to-one mapping between the input features $f'$ and the latent variables $z$. This is also the guarantee for our method: it prevents representational collapse and ensures that minimizing our alignment loss, $\mathcal{L}_{\text{align}}$, in the latent space has a direct and controllable effect on the alignment of the underlying feature distributions. The following Lemma formalizes this relationship.

**Lemma 1** (Bounded Feature Discrepancy). *Let $f'_{map}, f'_{track} \in \mathbb{R}^N$ be the compact feature inputs to the coupling layers $\mathcal{C}_{map}$ and $\mathcal{C}_{track}$. Assume their inverse transformations, $\mathcal{C}_{map}^{-1}$ and $\mathcal{C}_{track}^{-1}$, are L-Lipschitz continuous with a constant L Virmaux & Scaman (2018); Gouk et al. (2021). Then, the L2 distance between the input feature vectors is upper-bounded by a function of the alignment loss:*

$$\|f'_{map} - f'_{track}\|_2 \le L \cdot \sqrt{\mathcal{L}_{align}} + \delta, \tag{7}$$

*where $\delta$ is a constant representing the maximum structural discrepancy between the two inverse transformation functions over the domain of interest.*

*Proof.* The distance $\|f'_{map} - f'_{track}\|_2$ can be rewritten as $\|\mathcal{C}_{map}^{-1}(z_{map}) - \mathcal{C}_{track}^{-1}(z_{track})\|_2$. Applying the triangle inequality yields:

$$\|f'_{map} - f'_{track}\|_2 \le \|\mathcal{C}_{map}^{-1}(z_{map}) - \mathcal{C}_{map}^{-1}(z_{track})\|_2 + \|\mathcal{C}_{map}^{-1}(z_{track}) - \mathcal{C}_{track}^{-1}(z_{track})\|_2, \tag{8}$$

The first term is bounded by $L \cdot \|z_{map} - z_{track}\|_2$ due to L-Lipschitz continuity. Since $\|z_{map} - z_{track}\|_2 = \sqrt{\mathcal{L}_{align}}$, this term becomes $L \cdot \sqrt{\mathcal{L}_{align}}$. The second term reflects only the network structure discrepancy given the same input, thus is bounded by the maximum structural discrepancy $\delta$. Combining these terms gives the final result. $\square$

This Lemma provides a theoretical guarantee: minimizing our alignment loss $\mathcal{L}_{align}$ directly and provably tightens the upper bound on the distance between the input feature distributions.

### 3.4 PRACTICAL ANALYSIS

Building upon our theoretical guarantee, we provide a practical analysis to empirically verify the core mechanism of NexusFlow. We hypothesize that by explicitly aligning the latent feature distributions, our method forges a unified and more effective representation space for knowledge transfer. To validate this, we investigate two key properties of the learned representations: their **alignment** and **intrinsic dimensionality**. We assess alignment both qualitatively through t-SNE visualizations Maaten & Hinton (2008); Cai & Ma (2022) and quantitatively using the Maximum Mean Discrepancy (MMD) metric Gretton et al. (2012). We then analyze their intrinsic dimensionality via Principal Component Analysis (PCA) Abdi & Williams (2010); Del Giudice (2021) to ensure the unified representation is also complex enough to serve both disparate tasks.

**Distribution Alignment Analysis.** We first evaluate the alignment of the feature distributions. For a qualitative understanding, we employ t-SNE Maaten & Hinton (2008); Cai & Ma (2022) to visualize two sets of high-dimensional features: the intermediate features inside the task-formers, and the latent variables produced by our NexusFlow module. As illustrated in Figure 3, the visualization is revealing. The top row shows the intermediate features for tracking (blue) and mapping (red) form largely separate clusters. This is expected, as these features must retain task-specific information for the task-formers, but we observe that our NexusFlow brings the trend of clusters mixing. The critical distinction appears in the bottom row, which shows the latent variables after the surrogate networks.

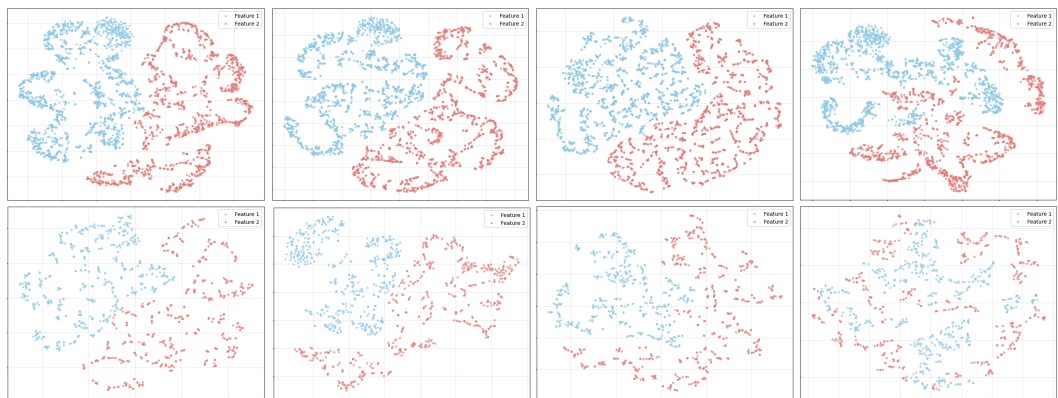

Figure 3: From left to right: t-SNE visualizations of Baseline, MTPSL, JTR, and NexusFlow(Ours). Top row shows the latent features from the two task-formers; bottom row shows those from the two coupling layers.

Here, the features from the baseline methods still exhibit a clear distributional shift. In contrast, the latent variables from our full NexusFlow model become substantially intermingled, providing visual evidence that our method creates a more unified manifold for knowledge transfer.

This qualitative observation is substantiated by our quantitative analysis using the Maximum Mean Discrepancy (MMD). MMD is a standard metric for measuring the distance between two distributions, where a lower score signifies greater similarity Gretton et al. (2012). We compute the MMD between the intermediate features from mapping-annotated samples and tracking-annotated samples. As shown in Table 1, NexusFlow module leads to a significant reduction in the MMD score compared to all baselines. This quantitatively validates that NexusFlow effectively reduces the distributional distance between the disparate tasks, creating the statistical foundation for knowledge transfer.

Table 1: MMD score comparison in same training setting (smaller means greater similarity).

|  | Ref | Baseline | MTPSL | JTR | Ours (w/o inv) | Ours |
|---|---|---|---|---|---|---|
| MMD | 0.23±0.32 | 2.97±0.54 | 2.81±0.36 | 2.77±0.54 | 2.54±0.29 | 1.56±0.47 |

**Intrinsic Dimensionality Analysis.** Beyond showing the distributions are closer, we analyze the feature distributions' intrinsic complexity using Principal Component Analysis (PCA). Our analysis is guided by the premise that a representation with a slower decay in its eigenvalue magnitudes holds more informative dimensions, making it more suitable for complex, multi-tasks Goel & Klivans (2017); Ansuini et al. (2019); Kim et al. (2023). As shown in the Scree plot Zhu & Ghodsi (2006) in Figure 4, the features from the baseline and a variant of our model where we ablate the invertible layer ('Ours (w/o inv)') exhibit rapid eigenvalue decay, suggesting their representations are more com-

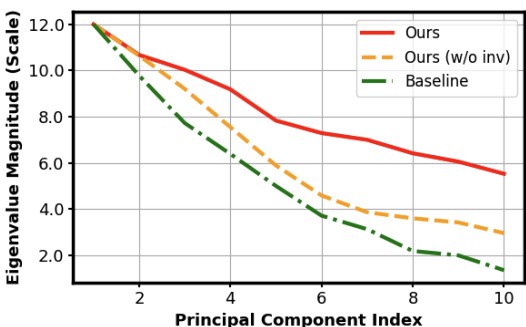

Figure 4: Figure of eigenvalue magnitudes decay.

pressible and may lose nuanced information. In contrast, the eigenvalue of our full NexusFlow model decays at a slower rate. This provides evidence that our alignment process forges a more complex and information-rich feature space, one that retains the high dimensionality necessary to effectively serve two structurally disparate tasks.

## 4 EXPERIMENTS

**Dataset.** We conduct our experiments on nuScenes Caesar et al. (2020), a large-scale and challenging public benchmark for autonomous driving. The dataset contains 1000 driving scenes, captured in

| Method | Multi-object Tracking | | | | Online Mapping | | | |
|---|---|---|---|---|---|---|---|---|
| | AMOTA↑ | AMOTP↓ | Recall↑ | IDS↓ | Lanes↑ | Drivable↑ | Divider↑ | Crossing↑ |
| Oracle (fully supervised) | 0.323 | 1.328 | 0.431 | 696 | 31.4 | 71.42 | 34.4 | 21.3 |
| Baseline Hu et al. (2023) | 0.289 | 1.488 | 0.362 | 1025 | 27.1 | 62.7 | 22.6 | 14.1 |
| MTPSL Li et al. (2022) | 0.255 | 1.504 | 0.321 | 1089 | 27.0 | 59.6 | 21.7 | 11.5 |
| JTR Nishi et al. (2024) | 0.197 | 1.547 | 0.317 | 774 | 25.1 | 57.6 | 21.0 | 12.1 |
| **Ours (One stage)** | 0.318 | 1.353 | 0.407 | 734 | 37.0 | 63.7 | 29.4 | 21.5 |
| **Ours (Two stage)** | **0.329** | **1.322** | **0.428** | **690** | **37.1** | **64.5** | **30.0** | **22.8** |

Table 2: **Multi-object tracking and Online mapping results.** Ours achieves competitive performance against Sota methods on both tasks. For online mapping, we report segmentation IoU (%).

Boston and Singapore, which are officially divided into 700 for training, 150 for validation, and 150 for testing. In total, it comprises approximately 40,000 annotated keyframes. Moreover, nuScenes provides rich annotations for our two disparate tasks: over 1.4 million 3D object bounding boxes for tracking and detailed, city-level vectorized maps for map reconstruction. For vision-centric methods, it provides the standard sensory input, which includes images from six surrounding cameras, intrinsic and extrinsic calibration matrices, and vehicle ego-motion data Zheng et al. (2023); Liu et al. (2024); Chen et al. (2024).

**Experimental Setup.** To ensure a fair comparison, all methods are trained from scratch by using the same hyperparameters, including learning rates, batch sizes, and loss balancing factors Lin et al. (2023). All experiments are conducted with four NVIDIA A100 80G GPUs. To rigorously evaluate performance in the partially supervised setting with an explicit domain gap, we design a specific data protocol based on the data locations in nuScenes. Specifically, we provide ground-truth annotations for the **mapping** task *only* for scenes in **Boston**. Conversely, ground-truth annotations for the **multi-object tracking** task are provided *only* for scenes in **Singapore**. This challenging, geographic domain-based partial supervision protocol is applied consistently across all evaluated methods, as well as in our analysis and ablation studies. For an upper-bound comparison, we also train an **Oracle** model that is fully supervised with all mapping and tracking labels from both cities.

**Baseline Architecture.** To provide a strong and consistent foundation for our experiments, we adopt the SOTA **UniAD** Hu et al. (2023) architecture as the backbone for all evaluated methods. This includes the fully supervised **Oracle**, and the standard partially supervised **Baseline**. To show the superiority of our method, we re-implement two PS-MTL methods, **MTPSL** Li et al. (2022) and **JTR** Nishi et al. (2024). Although originally designed for dense prediction tasks, we carefully adapt their core knowledge transfer mechanisms and integrate them into the UniAD architecture. This ensures a fair and controlled comparison where the primary difference between methods is the specific strategy used for partially supervised learning, not the underlying perception model.

**Quantitative Results.** We present our results in Table 2, with the main metric for each task highlighted in gray. While training is strictly conducted under our scenario-based partial supervision protocol, evaluation is performed on the complete validation set, including scenes from both Boston and Singapore. Our proposed NexusFlow significantly outperforms all partially supervised baselines and, remarkably, achieves performance competitive with the fully supervised Oracle.

- *Multi-Object Tracking:* For the tracking task (left side of the table), NexusFlow sets a new SOTA for this challenging PS-MTL setting. It surpasses the strong JTR Nishi et al. (2024) baseline by a large margin of **+13.8% AMOTA** and improves upon the standard Baseline Hu et al. (2023) by **+4.0% AMOTA**. Furthermore, our method achieves the lowest ID Switch score, demonstrating a superior ability to maintain temporal consistency for each tracklet.
- *Online Mapping:* For the mapping task (right side of the table), our method shows substantial gains in segmenting crucial road elements. Notably, it outperforms the Baseline by over **+10% IoU** on lanes, a critical component for safe downstream motion planning. This result confirms that effective knowledge was transferred from the tracking-annotated domain (Singapore) to enhance the mapping performance.

**Qualitative Results.** These quantitative results are corroborated by our qualitative analysis in Figure 5. We observe that our model is able to detect objects more accurately. A full video comparison is provided in the supplementary material.

Baseline method

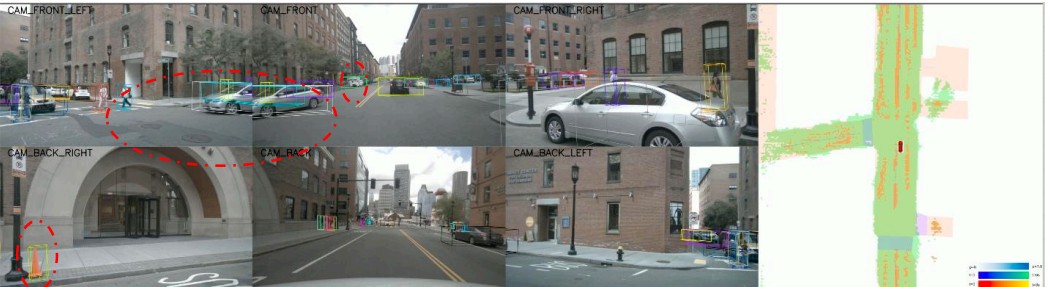

Ours

Figure 5: **Visualization results.** We show results for multi-object tracking and online mapping tasks in surround-view images and BEV from left to right. We highlight the difference with red dashed circle.

**Ablation Study.** We conduct an ablation study to validate our design choices, with results presented in Table 3. We compare our full model against two variants: the **'Baseline'** (without the NexusFlow module) and **'Ours (w/o inv)'** (without the invertible coupling layers). The results confirm that our full model significantly outperforms both variants. While 'Ours (w/o inv)' surpasses the 'Baseline', the performance gap to the full model underscores the critical role of the invertible transformation. Furthermore, we find that a coupling layer depth of 6 achieves the optimal performance, which we adopt in our final design.

| Method | Multi-object Tracking | | | | Online Mapping (IoU %) | | | |
|---|---|---|---|---|---|---|---|---|
| | AMOTA↑ | AMOTP↓ | Recall↑ | IDS↓ | Lanes↑ | Drivable↑ | Divider↑ | Crossing↑ |
| Baseline Hu et al. (2023) | 0.289 | 1.488 | 0.362 | 1025 | 27.1 | 62.7 | 22.6 | 14.1 |
| Ours (w/o inv) | 0.214 | 1.507 | 1.355 | 731 | 32.3 | 56.8 | 27.7 | 21.9 |
| Ours (1 Layer) | 0.236 | 1.482 | 0.389 | 982 | 32.9 | 57.2 | 28.1 | 20.9 |
| Ours (2 Layers) | 0.258 | 1.475 | 0.396 | 913 | 33.5 | 58.2 | 28.3 | 21.0 |
| Ours (4 Layers) | 0.292 | 1.405 | 0.402 | 854 | 35.3 | 60.3 | 28.6 | 21.1 |
| **Ours (6 Layers)** | **0.329** | **1.322** | **0.428** | **690** | **37.1** | **64.5** | **30.0** | **22.8** |
| Ours (8 Layers) | 0.247 | 1.453 | 0.385 | 1054 | 33.1 | 57.9 | 28.5 | 20.7 |

Table 3: **Ablation study on multi-object tracking and online mapping.** Ours with 6 layers achieves the best performance on both tasks.

## 5 CONCLUSION

In this paper, we tackled the challenging and previously unexplored problem of Partially Supervised Multi-Task Learning (PS-MTL) for structurally disparate tasks, focusing on the complex autonomous driving setting of joint map reconstruction and multi-object tracking. We introduced **NexusFlow**, a novel, lightweight, and plug-and-play framework that effectively aligns the latent feature distributions of these heterogeneous tasks. Through theoretical and practical analysis, we demonstrated that our invertible, flow-based alignment provides a principled mechanism for knowledge transfer, enabling our method to establish a new SOTA on the challenging, domain-shifted nuScenes dataset.

**Ethics Statement.** This work does not involve sensitive personal data, or proprietary information. All experiments were conducted in servers using publicly available datasets and models under appropriate licenses. The authors affirm compliance with the ICLR Code of Ethics and uphold the principles of scientific integrity, transparency, and responsible stewardship.

**Reproducibility Statement.** We have taken extensive measures to ensure the reproducibility of our results. Critical implementation details, model configurations, and experimental settings are described in the main paper. Our code and demo video are available in the supplementary.

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

# APPENDIX

## A   LLM USAGE STATEMENT

Large Language Models (LLMs) were not used to generate, analyze, or create any of the content, results, or figures presented in this paper. LLMs were only employed after the full manuscript was completed, and solely for light editing of grammar and phrasing. All scientific ideas, experimental design, implementation, and writing were conducted entirely by the authors.

