# OpenReview forum: "NexusFlow: Unifying Disparate Tasks under Partial Supervision via Invertible Flow Networks"
_ICLR.cc/2026/Conference — ICLR 2026 Conference Withdrawn Submission_

### Official Review · Reviewer_Nsho · 2025-10-24

**Soundness:** 3
**Presentation:** 2
**Contribution:** 3
**Rating:** 8
**Confidence:** 3

**Summary:**

This paper proposes a method of multi-task learning applied to the unusual, and challenging, situation where the tasks use different forms of annotation and where the annotations for different tasks are only provided for disjoint sub-sets of the training data (an extreme version of Partially Supervised Multi-Task Learning). Furthermore, the paper proposes a method to address this problem which is shown to produce good results.

**Strengths:**

The paper proposes a new problem within the domain of Partially Supervised Multi-Task Learning, which is highly relevant to practical applications such as autonomous driving.

The proposed method produces strong performance.

**Weaknesses:**

The description of the proposed NexusFlow module is extremely brief and lacking in detail. In general, the methods are not described in sufficient detail to allow reproduction of the results.

The paper only considers the situation where there are only two tasks, and there is not a discussion as to how the method could be applied to more than two tasks.

Figure 2 is unclear and fails to help explain the proposed method.

**Questions:**

How large are the fixed-dimension embeddings (l.240) extracted from each task former? Which intermediate features are used to generate these embeddings? How sensitive are the results to the choice of intermediate features and the dimensionality of the embedding?

What is the reason for choosing a Deformable DETR (l.241) as the Feature Aggregator? Could alternative architectures be used? How sensitive are the results to this choice?

How large are the MLPs implementing the coupling layer (l.250).

The paper refers the reader to another paper for all details of the training set-up (l.399). However, if this is a novel method how can all such information be published? What values were used for the five separate loss weights used in the proposed method (eqs. 2, 3, and 6)? How were these values selected? How sensitive are the results to these choices?

How would the proposed method scale to more than two tasks? Would a separate NexusFlow be required for each pair? How would the number of parameters/computational cost increase with the number of tasks? Presumably more tasks would also require more hyper-parameters to be tuned (such as the loss weights). How could hyper-parameter tuning be performed efficiently?

---

### Official Review · Reviewer_GB5e · 2025-10-27

**Soundness:** 2
**Presentation:** 2
**Contribution:** 3
**Rating:** 4
**Confidence:** 4

**Summary:**

>This paper presents NexusFlow, a plug-and-play framework designed to tackle the critical challenge of Partially Supervised Multi-Task Learning (PS-MTL) involving structurally disparate tasks under domain-partitioned supervision. Existing PS-MTL approaches have traditionally focused on simpler settings with homogeneous tasks, leaving the more realistic and complex scenario of learning from heterogeneous tasks with distinct geographic supervision domains largely unexplored. NexusFlow addresses this by introducing independent invertible coupling layers within surrogate networks to explicitly align the latent feature distributions of the disparate tasks—specifically, dense map reconstruction and sparse multi-object tracking in autonomous driving—creating a unified, information-preserving representation. The framework is validated on the challenging nuScenes benchmark, demonstrating state-of-the-art results that significantly outperform strong partially supervised baselines.

**Strengths:**

>S1. The proposed technique for scalable Partially Supervised Multi-Task Learning in the context of structurally disparate tasks is highly practical and relevant for real-world applications.

>S2. The proposed method appears to be simple yet efficient. Its plug-and-play nature suggests it is largely model-agnostic and potentially applicable to a wide variety of network architectures.

>S3. The method achieves a significant performance improvement in the specific 2-task PS-MTL scenario evaluated for Autonomous Driving.

**Weaknesses:**

>W1. The evaluation is currently restricted to only one dataset, one specific 2-task PS-MTL scenario, and only the UniAD architecture. This limited scope severely restricts the verification of the proposed method's generalization capability. To substantiate the claim of general applicability, a more comprehensive evaluation is necessary, testing the method across at least one of the following: additional datasets, different MTL scenarios, or alternative architectures.

>W2. The title of the paper appears to misrepresent the scope of the paper. Since the evaluation is confined exclusively to autonomous driving benchmarks, the claim of presenting a general PS-MTL technique seems overstated, as it is only verified within this specific domain.

>W3. The choice of the invertible coupling layer remains unconvincing. The claim (`lines 102-106`) that "coupling layers provide a bijective mapping that preserves all task-relevant information while enabling flexible transformation... maintaining information capacity" lacks sufficient theoretical reference within the text. While the Intrinsic Dimensionality Analysis provides empirical support for the distribution alignment, the lack of theoretical justification, combined with the provided MMD experiments (which only support distribution distance), makes the rationale somewhat unconvincing.

>W4. The purpose of Lemma 1 is unclear. It seems that the provable tightening of the upper bound on the input feature distance would be guaranteed even if a simple MSE loss were applied to the non-coupling features ($f_{map}', f_{track}'$), making the specific contribution of the coupling layer trivial. Also, this proof does not serve as a sufficient theoretical justification for the choice of the invertible coupling layer.

>W5. The foundational work proposing the invertible coupling layer, RealNVP [1], is not cited in either the Related Work or Method sections, with only subsequent works being referenced.

### __*References*__
_[1]. Dinh, Laurent, Jascha Sohl-Dickstein, and Samy Bengio. "Density estimation using real nvp." arXiv preprint arXiv:1605.08803 (2016)._

**Questions:**

>Q1. Could the authors provide a theoretical reference or more rigorous analysis to support the key claim (lines 102-106) that the invertible coupling layer actively preserves all task-relevant information and maintains information capacity under drastic structural differences?

>Q2. Why was the non-invertible projection layer $g'(\cdot)$ excluded from the theoretical analysis in Lemma 1, considering its crucial role in the overall feature transformation pipeline?

---

### Official Review · Reviewer_oHjP · 2025-11-01

**Soundness:** 2
**Presentation:** 2
**Contribution:** 2
**Rating:** 2
**Confidence:** 3

**Summary:**

This work studies the problem of partially supervised multi-task learning (PS-MTL) for structurally disparate tasks under domain-partitioned supervision. Specifically, it focuses on multi-object tracking (sparse, instance-level set) and map reconstruction (dense, grid-based representation) tasks with supervision split across geographically distinct domains. It proposes NexusFlow, a lightweight framework that uses independent invertible coupling layers to align the task-specific feature distributions for effective cross-task knowledge transfer. Experiments on the nuScenes dataset show benefits over alternative PS-MTL strategies for knowledge transfer (MTPSL, JTR), with competitive performance to a fully supervised oracle.

**Strengths:**

- PS-MTL in the context of structurally disparate tasks under domain-partitioned supervision is a relevant problem since labels are often collected in different geographic domains for different types of tasks.
- The proposed framework uses insights from invertible flow-based models to align the latent feature distributions of different tasks.
- Experiments on the nuScenes dataset (Tab.2) show benefits over alternative PS-MTL strategies (MTPSL, JTR).
- Distribution alignment and intrinsic dimensionality analysis in Sec.3.4 provide insights into the learned representations.

**Weaknesses:**

- The text emphasizes NexusFlow as a plug-and-play framework (L017, L094, L118, L253 and more). To validate this, it would be useful to show results on at least one more baseline (other than UniAD). For example, GenAD (L162: Zheng et al. 2024, code is publicly available) also makes predictions for both sparse (detection, trajectory prediction) and dense (mapping) tasks. NexusFlow can be applied to GenAD as well and compared with other PS-MTL strategies (MTPSL, JTR). This would help show that the gains are not specific to UniAD but can be applied to other models as well.
- It'd be useful to have a simple finetuning baseline without the feature alignment loss in Eq.6. For example, the non-linear projections of the task-specific features (via a learned network) can be concatenated and then used for multi-object tracking and map reconstruction tasks. This would help understand if the proposed approach is indeed more effective than simply fusing intermediate task-specific features and finetuning with additional network capacity.
- Tab.1 shows that applying existing PS-MTL strategies (MTPSL, JTR) to the baseline model leads to drops in performance. Ideally, these strategies should help with knowledge transfer and at least not degrade performance. Is there any insight into why these strategies do not work when added to baseline in this setting?
- It is interesting that the performance with NexusFlow is competitive with the full supervised oracle in Tab.2 since the oracle should have access to much more annotations. Is the label distribution skewed towards one domain for one task and the other domain for the other task? Currently, mapping labels are used for Boston and tracking labels for Singapore (L401-403). If the labels are reversed (mapping labels for Singapore and tracking labels for Boston), does this trend still hold? This would further strengthen the argument about the effectiveness of NexusFlow in domain-partitioned supervision.
- In the ablation study (L462-463), is the NexusFlow variant without the invertible coupling layer equivalent to a simple finetuning baseline? Or are there any other major differences? This variant improves performance on most metrics (except AMOTA, AMOTP in Tab.3). This seems to indicate that invertible coupling layers are not essential for knowledge transfer. It'd be useful to clarify this.
- Is Fig.2 just for illustration purposes, or is the information sharing aspect between the two tasks related to feature visualizations in Fig.3?

**Questions:**

There are several concerns about the experiments (more details in the weaknesses above):
- To validate the plug-and-play aspect, the gains should be evident on at least one more model.
- A simple finetuning baseline is required to verify that the proposed approach is indeed more effective than simpler alternatives.
- Several clarifications are required about the performance of MTPSL, JTR, comparison to the fully supervised oracle, and ablations.

---

### Official Review · Reviewer_tsFY · 2025-11-04

**Soundness:** 3
**Presentation:** 2
**Contribution:** 3
**Rating:** 6
**Confidence:** 3

**Summary:**

This paper tackles the challenging problem of Partially Supervised Multi-Task Learning (PS-MTL), focusing on a setting that has been unexplored. The authors identify a critical gap in existing literature, which has primarily focused on homogeneous, dense prediction tasks (e.g., segmentation and depth). The paper's primary contribution is to formulate and address PS-MTL for structurally disparate tasks specifically, dense map reconstruction and sparse multi-object tracking.To make this problem more realistic and challenging, the authors introduce a new benchmark setting based on domain-partitioned supervision. Instead of assuming random label dropout, this setting mimics a practical data collection scenario where supervision for each task comes from different geographic domains (e.g., mapping labels only from Boston, tracking labels only from Singapore).

To solve this, the paper proposes NexusFlow, a lightweight, plug-and-play alignment module. NexusFlow introduces a pair of surrogate networks that are input into the task-specific decoders (e.g., MapFormer and TrackFormer). The core idea is the use of invertible coupling layers, a component inspired from flow-based models. These layers project intermediate features from each task into a canonical latent space. A simple L2 alignment loss, is then applied to pull these latent representations together, creating a unified feature space and enabling knowledge transfer. The authors argue that the invertibility of these layers is key, as it provides a bijective mapping that can align distributions without the "representational collapse"  that might otherwise occur when forcing features from such disparate tasks to be similar. This claim is supported by both a theoretical justification (Lemma 1), which bounds the feature-space distance by the latent-space loss, and a strong practical analysis. This analysis uses t-SNE, MMD, and PCA to empirically demonstrate that NexusFlow achieves superior feature alignment (lower MMD) and learn complex, information-rich representation (slower eigenvalue decay).

Experiments are conducted on the nuScenes dataset using the UniAD architecture as a backbone. Under their domain-partitioned protocol, NexusFlow significantly outperforms the baseline UniAD model and other PS-MTL methods (MTPSL, JTR). Notably, it achieves a +13.8\% AMOTA gain in tracking and a +10\% IoU gain in lane mapping, demonstrating effective knowledge transfer across disparate tasks and domains.

**Strengths:**

The following are some strengths of this work:
- The paper's primary strength is its formalization of a  challenging problem. The introduction of "domain-partitioned supervision" is interesting, as it reflects a far more realistic scenario than the standard "random dropout" assumption, combining task heterogeneity with a domain gap.
- The choice of invertible coupling layers is well-motivated. The authors provide a clear and compelling argument for why invertibility is necessary: it enables alignment via a bijective mapping, which preserves information and prevents the latent representation from collapsing. This is a crucial property when dealing with tasks as different as sparse tracking and dense mapping.
- The t-SNE visualizations and, more importantly, the quantitative MMD scores  clearly demonstrate that NexusFlow achieves a more unified latent distribution compared to baselines.
- The PCA scree plot analysis showing a slower eigenvalue decay is quite interesting. The authors provide evidence for their claim: their method avoids representational collapse and creates a "more complex and information-rich feature space"  capable of supporting both disparate tasks.

**Weaknesses:**

Im not an expert on Multi-Task Learning, so I shall defer the technical aspects of the weakness to my fellow reviewer colleagues. However, I have the following weaknesses from reading about the work:
- The paper's core premise is aligning "structurally disparate" tasks (dense map vs. sparse tracks). However, the technical implementation seems different. The NexusFlow module does not operate on the disparate outputs themselves. Instead, it operates on intermediate features that have already been passed through a "Feature Aggregator" ($g^{\prime}(\cdot)$), which is a deformable transformer that compresses them into a "fixed-dimension embedding" ($f_{map}^{\prime}$ and $f_{track}^{\prime}$). This means the invertible layers are aligning two already-homogenized fixed-dimension vectors, not two structurally disparate representations.
- I find the results in Table 2 to be counter-intuitive. The proposed method ("Ours (Two stage)") achieves better performance on several mapping metrics than the "Oracle (fully supervised)" model. Lanes IoU: Ours (37.1) vs. Oracle (31.4) and Crossing IoU: Ours (22.8) vs. Oracle (21.3). It implies that training with less data (the partial, domain-partitioned dataset) leads to a better model than training with all data (full supervision in both domains). This could suggest that the full supervision setting suffers from some kind of catastrophic forgetting or it could be a statistical anomaly or a flaw in the Oracle's training.
- The paper's problem formulation states that each sample \(x_i\) has labels for only one task, from only one domain. However, the alignment loss is defined as $L_{\text{align}} = \|| z_{\text{map}} - z_{\text{track}} \||_2$

(assuming a typo correction), where $z_{\text{map}}$ and $z_{\text{track}}$  are derived from $f_{\text{map}}^{\prime}$ and $f_{\text{track}}^{\prime}$, respectively.
This loss seems to require both $z_{\text{map}}$ and $z_{\text{track}}$ for a given sample. Could the authors please clarify how this is computed?

**Questions:**

Please refer to the weakness section

---

### Note · Authors · 2025-11-13

I have read and agree with the venue's withdrawal policy on behalf of myself and my co-authors.